# ZnO Composite Graphene Coating Micro-Fiber Interferometer for Ultraviolet Detection

**DOI:** 10.3390/s20051478

**Published:** 2020-03-08

**Authors:** Tao Shen, Xiaoshuang Dai, Daqing Zhang, Wenkang Wang, Yue Feng

**Affiliations:** 1Heilongjiang Provincial Key Laboratory of Quantum Manipulation & Control, Harbin University of Science and Technology, Harbin 150080, China; taoshenchina@163.com (T.S.); 13103809866@163.com (X.D.); 2Key Laboratory of Engineering Dielectrics and Its Application, Ministry of Education, Harbin University of Science and Technology, Harbin 150080, China; beirut_silence@163.com (D.Z.); wenk1227@163.com (W.W.)

**Keywords:** ZnO composite graphene, interferometer, optical fiber sensing, ultraviolet sensor

## Abstract

A simple and reliable ultraviolet sensing method with high sensitivity is proposed. ZnO and ZnO composite graphene are successfully prepared by the hydrothermal method. The optical fiber sensor is fabricated by coating the single-mode-taper multimode-single-mode (STMS) with different shapes of ZnO. The effects of the sensitivity of ultraviolet sensors are further investigated. The results show that the sensor with ZnO nanosheets exhibits a higher sensitivity of 357.85 pm/nW·cm^−2^ for ultraviolet sensing ranging from 0 to 4 nW/cm^2^. The ultraviolet characteristic of STMS coated flake ZnO composite graphene has been demonstrated with a sensitivity of 427.76 pm/nW·cm^−2^. The combination of sensitive materials and optical fiber sensing technology provides a novel and convenient platform for ultraviolet detection technology.

## 1. Introduction

Ultraviolet detection technology has a wide range of applications. It is not only not susceptible to long-wave electromagnetic interference, working in a strong electromagnetic radiation environment, but also has good concealment. This technology has been widely used in military, communication, industry and other fields [1,2,3,4]. In the past, the photomultiplier, Si-based ultraviolet photodetectors are the main ways to achieve ultraviolet detection [5,6]. However, the photomultiplier has the disadvantages of high power consumption, large size. In addition, it can only work in low-temperature and high-pressure environments. Si-based ultraviolet photodetectors have low quantum efficiency and require expensive filters to suppress interference. With the development of science and technology, ZnO-based ultraviolet photodetectors have injected new vitality into ultraviolet detection technology [7,8].

In recent years, ZnO-based ultraviolet detection technology has been widely reported. It has the advantages of high responsiveness, high quantum efficiency, high response speed and good signal-to-noise ratio [9,10]. ZnO is a group II-VI wide-bandgap semiconductor material with a direct bandgap of 3.37 eV. Due to its inherent characteristics, including high exciton binding energy, high electron mobility, high radiation hardness and compliance with conventional etching, ZnO has good sensitivity and appropriate properties for ultraviolet radiation detection. Feng et al. [11] reported an optical testing method for an ultraviolet (UV) sensor. The fabrication of a UV-sensing element consists of a tapered micro-nano fiber coated with ZnO nanorods. The output optical power changes are detected in response to UV irradiation intensity. Sahatiya et al. [12] investigated a simple, one-step by electrospinning across electrodes and its subsequent use for ultraviolet (UV) detection. What is more, the morphology of ZnO has a great influence on the physical and chemical properties of the material [13,14]. As a new two-dimensional material, graphene has a large specific surface area and stable electrical conductivity, making it one of the best choices for metal oxide carrier materials [15]. It not only improves the electronic transmission performance of metal oxide materials, but also effectively prevents the agglomeration of metal oxide nanoparticles to some certain extent. The synergistic effect of graphene and ZnO can further enhance its ultraviolet detection performance. The applications of ZnO composite graphene in ultraviolet detection have also been studied [12,16].

At present, most ultraviolet detection technologies convert optical signals into electrical ones. What is different from the past is that a novel and simple method for ultraviolet detection is proposed in this paper. Optical fiber sensing technology is combined with ultraviolet sensitive material to explore their ultraviolet detection performance. In this paper, the fiber structure of single-mode-taper multimode-single-mode (STMS) is combined with ZnO and ZnO composite graphene to determine their sensitivity to ultraviolet by studying the shift of interference spectra.

## 2. Sensor Theoretical Analysis

### 2.1. Structure of the Sensors

Based on the principle of the evanescent field, the sensor of STMS coated ZnO or ZnO composite graphene is designed. The length of multimode fiber is 8 cm, fiber core diameter is 62.5 μm. After preliminary experiments by different sizes of multimode micro-nano fiber, the taper multimode fiber is further modified with a diameter of 4 μm. The schematic diagram of the sensor is shown in Figure 1. The experimental devices consist of a magnified spontaneous emission (ASE) source, the ultraviolet sensor and an optical spectrum analyzer (OSA). The central wavelength of the ASE source is 1550 nm, and the wavelength range is from 1523 nm to 1573 nm. The resolution of OSA is 0.02 nm and the detection range is from 600 nm to 1700 nm. The ultraviolet sensor consists of optical fiber structure and sensitive materials. The optical fiber structure is composed of two single-mode fibers and one tapered multimode fiber. The tapered multimode fiber is in the middle part. The sensitive material is different shapes of ZnO and flakes ZnO composite graphene. The ultraviolet characteristics of the sensors are tested by exposing the sensors in the ultraviolet radiation luminance from 0 to 4.164 nW/cm^2^. 

### 2.2. Operation Principle

The light is transmitted only as the base mode in a single-mode fiber. However, when it is transmitted to the fusion interface between the single-mode fiber and the tapered multimode fiber, multiple higher-order modes are excited in the tapered multimode fiber. These modes will propagate with different propagation constants along the tapered multimode fiber to the waist of the fiber, further to the interface of its fusion with the output single-mode fiber, and interfering with the base mode to re-enter the single-mode fiber. For the excited higher-order mode, the electric field component is expressed as:(1)$$E\left(r,z\right)=\sum_{m=1}^{M}c_{m}E_{m}\left(r\right)\exp\left(i\beta_{m}z\right)$$
where *E_m_*(*r*) is the distribution of electric field, *β_m_* is the propagation constant of the longitudinal m-order mode, and *c_m_* is the mode efficiency of excitation from the base mode of single-mode fiber to the tapered multimode fiber. The power of higher-order mode in the tapered multimode fiber is closely related to the coupling coefficient *η_m_*
(2)$$\eta_{m}=c_{m}^{2}$$

The larger *η_m_*, the higher the power of the higher-order mode. Under the weak derivative approximation, the difference of the higher-order mode propagation constants in the tapered multimode fiber can be expressed as [17,18]:(3)$$\beta_{m}-\beta_{n}=\frac{\mu_{m}^{2}-\mu_{n}^{2}}{2ka^{2}n_{T}}$$
where *n_T_* is the refractive index of the tapered multimode fiber, *a* is the beam waist radius of the tapered multimode fiber and *m* and *n* represent different modes. Besides, *μ**_m_* and *μ**_n_* are defined as $\mu_{m}=\pi (m-1/4)$ and $\mu_{n}=\pi (n-1/4)$. When the length of the sensing region is *L*, the phase difference is as follows:(4)$$\Delta \varphi =\left(\beta_{m}-\beta_{n}\right)L=\frac{\lambda \left(\mu_{m}^{2}-\mu_{n}^{2}\right)}{4\pi a^{2}n_{T}}\cdot L$$

According to the light interference theory, the transmitted light will be coherently superimposed when the phase difference between two different modes of light satisfies the following expression:(5)$$\left(\beta_{m}-\beta_{n}\right)L=2\pi N$$

We can obtain the wavelengths at which maximum interferences happen, that is [19]:(6)$$\lambda_{c}=\frac{16n_{T}a^{2}N}{\left(m-n\right)\left[2\left(m+n\right)-1\right]L}$$

When the sensitive material coated on the outer surface of the tapered multimode fiber is affected by external ultraviolet radiation, its refractive index will change rapidly, causing the propagation constant and mode field distribution in the tapered multimode fiber to change. Furthermore, the characteristic wavelength of the interference spectrum shifts.

## 3. Experiments and Results

### 3.1. Ultraviolet Sensing Characteristics of STMS Coated ZnO and ZnO Composite Graphene

#### 3.1.1. Preparation of ZnO and ZnO Composite Graphene

ZnO microspheres were prepared by the sol–gel assisted hydrothermal method. Firstly, 5.0 g C_6_H_8_O_7_, 3.5 g Zn(NO_3_)_2_·6H_2_O and 100 mL deionized water stir at 80 °C for 2 h. After that, transfer to 100°C electric blast drying oven for 4 h. Then, NaOH solution was dropped into the mixture under stirring at room temperature. The pH of the suspension was adjusted to 13. The obtained suspension was transferred to the reaction kettle and heated at 120 °C for 17 h. The product was obtained and centrifuged at 6000 rpm/min for 10 min and, finally, rinsed with deionized water twice.

ZnO nanorods were prepared by the hydrothermal method. Firstly, 14.8 g Zn(NO_3_)_2_·6H_2_O and 100 mL deionized water were stirred at room temperature. NaOH solution was added until the PH was adjusted to 13. After ultrasound for 30 min, the solution as transferred to the reactor to join in the 50 mL deionized water 5 mL suspension. After ultrasonic for 30 min, the solution was transferred to the reaction kettle and heated at 200 °C for 25 h. The product was obtained and centrifuged with 6000 rpm/min for 10 min and, finally, rinsed with deionized water twice.

ZnO nanosheets were prepared by the hydrothermal method. Firstly, 1.0 g Zn(NO_3_)_2_·6H_2_O and 100 mL deionized water were stirred at room temperature for 1 h. NaOH solution was added until the PH was adjusted to 14. The obtained suspension was transferred to the reaction kettle and heated at 120 °C for 17 h. The product was obtained and centrifuged at 6000 rpm/min for 10 min and, finally, rinsed with deionized water twice.

Flake ZnO composite graphene was prepared by hydrothermal method. Firstly, 2.0 mg graphene powder was sonically dispersed in 15 mL deionized water for 1 h. Then, 0.5 g Zn(NO_3_)_2_·6H_2_O and 50 mL deionized water were mixed with ultrasonic dispersed graphene solution, and NaOH solution was dropped into the mixture under stirring at room temperature. The pH of the suspension was adjusted to 9.5. After that, the suspension was transferred to the reaction kettle and heated at 120 °C for 12 h. The product was obtained and centrifuged with 6000 rpm/min for 10 min. Finally, rinse with deionized water twice.

The morphology of the prepared ZnO and flake ZnO composite graphene was studied by SEM. The different shapes of ZnO are shown in Figure 2a–c. In Figure 2a, it can be seen that lots of spherical particles of ZnO agglomerate together. As shown in Figure 2b, ZnO is irregularly arranged in a rod-like structure. In Figure 2c, ZnO has a flaky structure. As shown in Figure 2d, it ZnO nanosheets are modified on the surface of graphene.

#### 3.1.2. Surface Functionalization of STMS 

The connection between the single-mode fiber and the tapered multimode fiber in the STMS structure was completed by a fiber fusion splicer. The STMS structure was fixed on a glass substrate and then washed with alcohol and deionized water to remove residual impurities. The STMS structure was placed in the electric blast drying oven with a temperature of 40 °C for 4 h. The prepared sensitive materials were dripped into the sensing area along the surface of the tapered multimode fiber. The entire structure was then placed into an electric blast drying oven with a temperature of 45 °C for 6 h, making sensitive materials tightly combined with tapered multimode fiber.

#### 3.1.3. Ultraviolet Sensing Experiment

The ultraviolet characteristics of the sensor were tested at room temperature (25 °C). In addition, the source that produces ultraviolet radiation was an LED ultraviolet lamp with a central wavelength of 365 nm, in which ultraviolet radiation luminance ranges from 0 to 4 nW/cm^2^. The distance between the LED lamp and sensing unit was constant. The output spectra for different ultraviolet radiation luminance were recorded using the OSA. On the one hand, the ultraviolet radiation luminance we adopted was from 0 to 4.164 nW/cm^2^, which is very small. On the other hand, the source that produced ultraviolet radiation was an LED ultraviolet lamp with a central wavelength of 365 nm. As we all know, ozone can be formed only under ultraviolet radiation with a wavelength shorter than 240 nm, because ultraviolet shorter than this wavelength can break the double bond of oxygen molecules to form ozone. Therefore, we did not need to consider the effect of ozone on the experimental results. Figure 3 shows the transmission spectra of ultraviolet sensors with different shapes of ZnO and flake ZnO composite graphene. During the exploration phase of the experiment, we found that it took 4 min for the spectrum to reach a stable state after each change in ultraviolet radiation illumination. Therefore, after 4 min of changing the ultraviolet radiation illumination, we recorded the spectrum in order to make the experimental data reliable. It can be seen from the Figure 3a, the STMS is coated ZnO microspheres, which shows a shift, and the sensitivity of sensor is 214.13 pm/nW·cm^−2^ when the selected wavelength is 1565 nm in the dark. After the sensor was coated with ZnO nanorods, the wavelength of the transmission spectra shifted, as shown in Figure 3b. The sensitivity of sensor was 162.13 pm/nW·cm^−2^ when the selected wavelength was 1541.68 nm in the dark. Figure 3c shows the transmission spectra of STMS coated ZnO nanosheets, in which sensitivity the of sensor is 357.85 pm/nW·cm^−2^ when the selected wavelength is 1547.47 nm in the dark. As for the STMS coated ZnO, the ultraviolet effect of ZnO nanosheets is stronger than that of the other two shapes. The change of ultraviolet radiation illuminance will cause the unbalanced carrier concentration in ZnO to increase, resulting in the change of the refractive index, and thus causing the shift of the transmission spectrum [20,21]. Due to the presence of flake ZnO composite graphene, taper multimode fiber can significantly enhance the surface evanescent fields that are extremely sensitive to the local refractive index. The strong interference can be observed in Figure 3d. The ultraviolet sensitivity is 427.76 pm/nW·cm^−2^ when the selected wavelength is 1556.87 nm in the dark. Graphene not only has a large specific surface area, but also effectively prevents agglomeration of ZnO particles during the preparation process. Therefore, graphene as the carrier of ZnO is an important way to improve its ultraviolet sensitivity. It is important to point out that the designed ultraviolet sensor does not respond to visible and infrared light.

The wavelength shifts of ultraviolet sensors versus the ultraviolet radiation luminance are shown in Figure 4, four linear fitting lines can be found in the range of whole ultraviolet radiation luminance measurement. The relevant results of STMS coated different shapes of ZnO and flake ZnO composite graphene are shown in Table 1. It can be found that the sensor with flake ZnO composite graphene shows a higher sensitivity to ultraviolet, which is 2.5 times that of coated ZnO nanorods. Because flake ZnO composite graphene has a specific surface area to contact with the external ultraviolet radiation luminance changes, the electron transfer increases and the refractive index also changes accordingly.

## 4. Conclusions

A high precision and sensitivity ultraviolet sensor based on STMS coated flake ZnO composite graphene is demonstrated. Different shapes of ZnO and flake ZnO composite graphene are synthesized via a facile hydrothermal method and the morphologies are also studied. Among STMS coated different shapes of ZnO, the sensor of STMS coated ZnO nanosheets shows a higher sensitivity, which is 357.85 pm/nW·cm^−2^. What is more, flake ZnO composite graphene exhibits additional synergistic effects and plays an important role in the interaction with the external ultraviolet environment, which is beneficial for ultraviolet sensing. The sensitivity of the sensor is 427.76 pm/nW·cm^−2^. The combination of sensitive materials and optical fiber sensing technology provides a novel and convenient platform for ultraviolet detection technology.

## Figures and Tables

**Figure 1 sensors-20-01478-f001:**
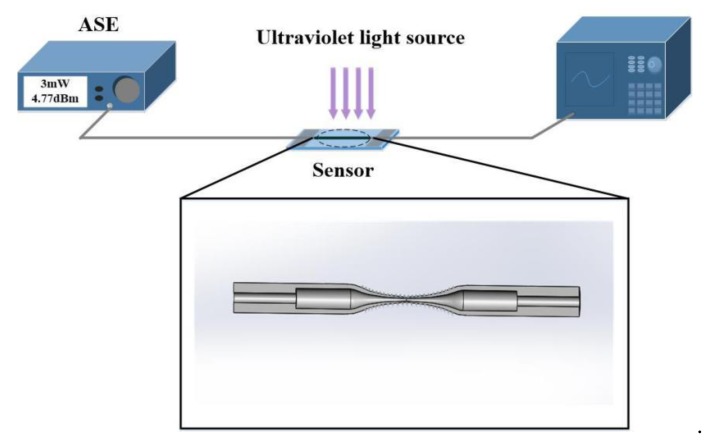
Schematic diagram of the experimental setup.

**Figure 2 sensors-20-01478-f002:**
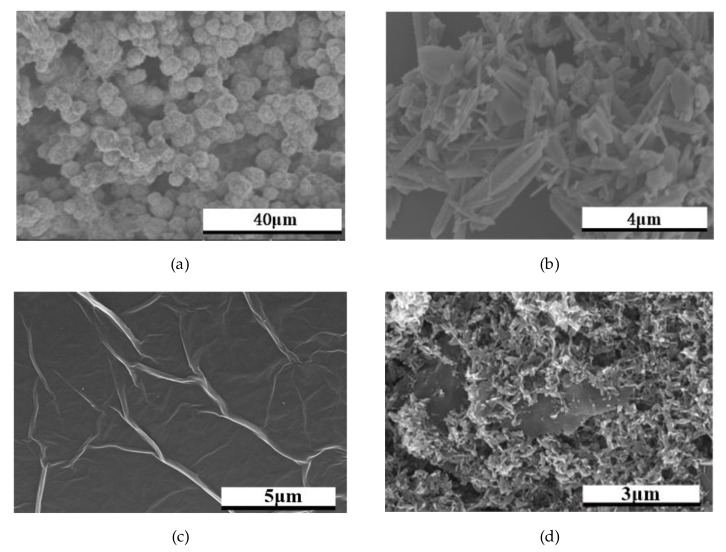
SEM image spectra for ZnO and ZnO composite graphene: (**a**) ZnO microspheres; (**b**) ZnO nanorods; (**c**) ZnO nanosheets; (**d**) flake ZnO composite graphene.

**Figure 3 sensors-20-01478-f003:**
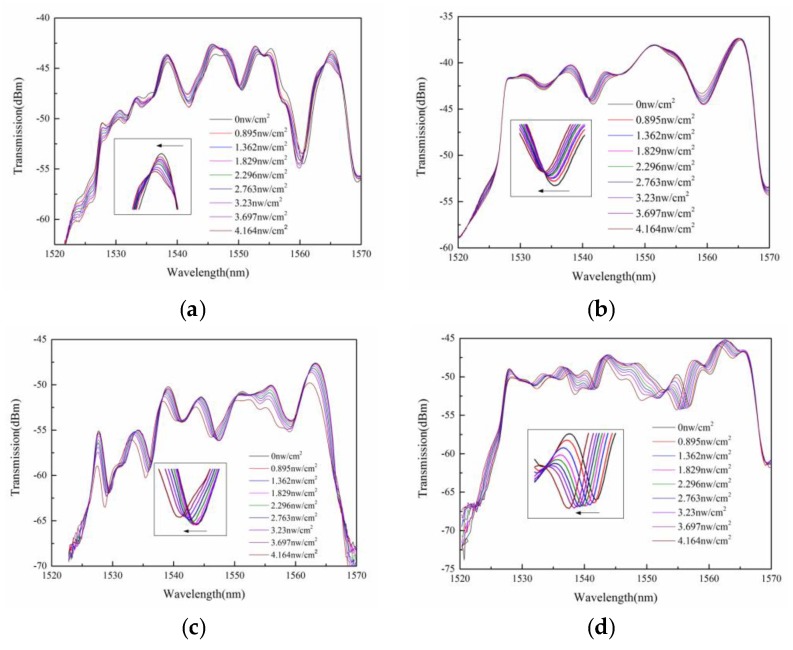
The transmission spectra of single-mode-taper multimode-single-mode (STMS) coated ZnO and ZnO composite graphene: (**a**) ZnO microspheres; (**b**) ZnO nanorods; (**c**) ZnO nanosheets; (**d**) flake ZnO composite graphene.

**Figure 4 sensors-20-01478-f004:**
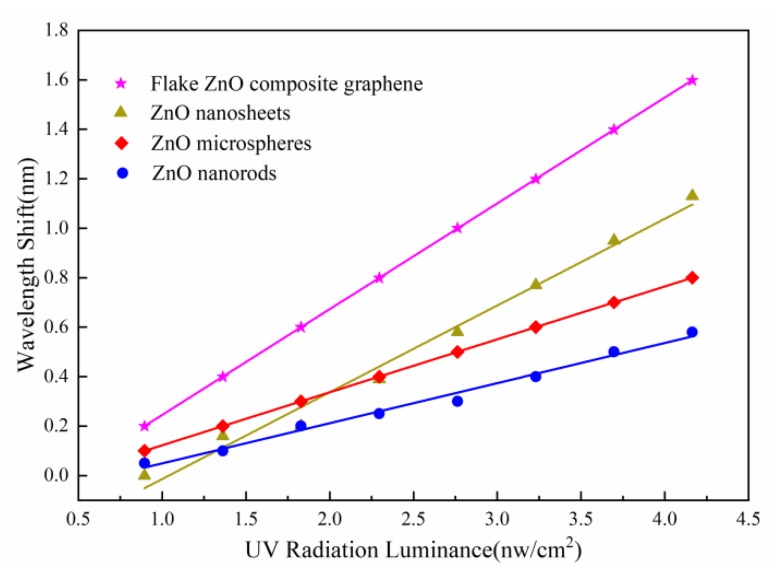
The wavelength shift versus ultraviolet radiation luminance for sensors.

**Table 1 sensors-20-01478-t001:** The results of the linear fitting of STMS coated ZnO and ZnO composite graphene.

Structure	Linear Fitting Line	Sensitivity/(pm/nW·cm^−2^)	R-Squared Value
STMS coated ZnO microspheres	*y*_0_ = 0.21413*x* − 0.09165	214.13	0.9999
STMS coated ZnO nanorods	*y*_1_ = 0.16213*x* −0.11261	162.13	0.9894
STMS coated ZnO nanosheets	*y*_2_ = 0.35785*x* − 0.37947	357.85	0.9865
STMS coated flake ZnO composite graphene	*y*_3_ = 0.42776*x*− 0.18276	427.76	0.9999

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
