# Peer review of "ZnO Composite Graphene Coating Micro-Fiber Interferometer for Ultraviolet Detection"

_sensors, 2020, doi:10.3390/s20051478_

Round 1
Reviewer 1 Report
The authors present a ZnO-based device for using in the single mode-taper multimode-single mode (STMS) for ultraviolet detection. Due to lack of information and experimental validation, discussed below, I suggest rejecting the article.
General points:
Please carefully revise the English grammar.
There is a lack of experimental details that must be improved. (e.g. sensor system, SEM, etc.)
The authors shall consider improving the introduction according to follow points:
Among other materials, is ZnO the best candidate to replace Si-based devices? Is there a commercial ZnO-based device for UV detection? What are the main challenges faced by researchers to replace Si by ZnO? ZnO-based devices are not sensitivity enough and requires the insertion of another material? As ZnO is widely studied as UV detector, please add a table or a paragraph discussing and comparing the advances in area. It is not clear why you are using STMS in this work. Is it superior to conventional methods?
Please check this information: “fiber core diameter is 62.5 cm”
ZnO is also highly sensitive to ozone gas produced by UV radiation, what care has been taken to ensure the efficiency of the proposed sensor?
Some coefficients of equations were not described, please verify.
Please check this information: “centrifuged with 6000 rmp/min”.
Please improve description of detection system. (parameters, equipment, fiber material, etc.)
ZnO preparation may be transferred to support information.
Please improve the scale bar readability from Fig. 2.
The response of unmodified optical fiber could be added to assess the ZnO surface modification.
A dimension nw/cm2 is presented in Figure 3. I understand that the authors are employing intensity as light parameter variation. So please correct it to nW/cm2.
Please explain how the spectra shift is important for your measurements. I would suggest you insert some arrows to indicate most important region to be considered in your analysis. The reader can also note that other peaks also varies with UV light intensity. Please explain why they were negligible.
The parameters values from equations presented in section 2.2 must be presented in the manuscript.
How many replicates were performed in the experiments? Please explain why no error bars are presented in figures.
There is no validation method to prove that proposed ZnO-based device has superior performance than Si-based devices.
Additional experiments with other wavelengths is required to assess the selectivity response of sensor.
There is no comparison of results with literature to prove the superiority of sensor.
Why the authors did not study ZnO and graphene composites with different ZnO structures, since it is expected that composite will present a higher performance than bare materials.
Reviewer 2 Report
Detailed comments are listed below.
* There is no pattern numbering in the publication.
* Line 70 - no definition of parameters in the fraction counter.
* What are the geometrical parameters of the STMS structure (e.g. a - beam waist radius?
* Figures 3a-3d are too small and illegible.
* The work would be more readable if the fragments of drawings were enlarged for which the wavelength shift was defined.
* No time dependence of the changes taking place. How often was the next transmission recorded?
* The nanowatt unit is incorrectly marked as nw. Should be nW.
Reviewer 3 Report
One major comment:
The introduction should highlight better the unknowns of this study: as it reads now the results seem just as-expected.
And a couple of minor ones:
Please increase the fonts in the figures, some of them are hardly readable.
The results of the liner fittings of Fig. 4 data should be reported in a Table instead of in the text, so to be easily seen and compared by the readers
Round 2
Reviewer 1 Report
The authors worked hard to improve the article according to the reviewers' suggestions, however some points still need to be verified as follow.
I shall insist to add a table or a paragraph comparing the results obtained with ZnO and ZnO/graphene composite with ones reported in literature.
This discussion must be added to the manuscript. “On the one hand, the ultraviolet radiation luminance we adopted is from from 0 to 4.164 nW/cm2, which is very small. On the other hand, the source that produces ultraviolet radiation is an LED ultraviolet lamp with a central wavelength of 365 nm. As we all know, ozone can be formed only under ultraviolet radiation with a wavelength shorter than 240 nm, because ultraviolet shorter than this wavelength can break the double bond of oxygen molecules to form ozone. Therefore, we do not need to consider the effect of ozone on the experimental results”
Please consider changing the color of scalebar and its caption of Figure 2 to black or white, because the red one is difficult to read.
Please, explain in the manuscript why the analysis of Figure 3 was performed considering only one peak shift and the others were not considered.
I totally agree that the experiments are repeatable. However, you shall demonstrate it in the manuscript using errors bar in your analytical curve. As you can observe in Figure 4, the fitting for ZnO nanosheets is not ideal and the first point of ZnO nanorods is missing.
About the selectivity experiments. In papers you mentioned below they present selectivity response experiments to other gases (e.g. methanol, alcohol, acetone, etc.). Since your proposal is a UV light sensor, you must demonstrate that your sensor do not respond to other lights (visible, infrared, etc.).
The paper "Zinc oxide nanoparticle incorporated graphene oxide as sensing coating for interferometric optical microfiber for ammonia gas detection" was published in Sensors and Actuators B in 2017. Besides, the paper "Fabrication of three-dimensional zinc oxide nanoflowers for high-sensitivity fiber-optic ammonia gas sensors" was published in applied optics in 2018.
Reviewer 2 Report
I present my comments (red)
2 Line 70 - no definition of parameters in the fraction counter.
Response: I am deeply sorry for such a stupid mistake due to my carelessness. The sentence of "The longitudinal propagation constants between two modes can be expressed as and μn" has been added. Thanks for the suggestion of the reviewer. μm
The explanation given in (lines 86 and 87) is completely incorrect. The authors of the work probably do not know what they are writing about. Parameters μn and μm are not propagation constants.
6) Comment: No time dependence of the changes taking place. How often was the next transmission recorded?
Response: We are very grateful for the reviewer's comments. During the exploration phase of the experiment, we found that it took 4 minutes for the spectrum to reach a stable state after each change in ultraviolet radiation illumination. Therefore, after 4 minutes of changing the ultraviolet radiation illumination, we recorded the spectrum in order to make the experimental data reliable.
The above explanation must be included in the work text. Information about the time between measurements is very important.
Other comments were taken into account by the authors of the work.
I have no more comment
